# Adaptive Spatial Transformation Networks for Periocular Recognition

**DOI:** 10.3390/s23052456

**Published:** 2023-02-23

**Authors:** Diana Laura Borza, Ehsan Yaghoubi, Simone Frintrop, Hugo Proença

**Affiliations:** 1Informatics Department, Faculty of Mathematics and Informatics, Babes Bolyai University, 1st Mihail Kogalniceanu Street, 400084 Cluj-Napoca, Romania; 2Department of Informatics, Hamburg University, 177 Mittelweg, 20148 Hamburg, Germany; 3IT: Instituto de Telecomunicações, University of Beira Interior, Marquês de Ávila e Bolama, 6201-001 Covilhã, Portugal

**Keywords:** periocular recognition, spatial transform, attention, biometrics

## Abstract

Periocular recognition has emerged as a particularly valuable biometric identification method in challenging scenarios, such as partially occluded faces due to COVID-19 protective masks masks, in which face recognition might not be applicable. This work presents a periocular recognition framework based on deep learning, which automatically localises and analyses the most important areas in the periocular region. The main idea is to derive several parallel local branches from a neural network architecture, which in a semi-supervised manner learn the most discriminative areas in the feature map and solve the identification problem solely upon the corresponding cues. Here, each local branch learns a transformation matrix that allows for basic geometrical transformations (cropping and scaling), which is used to select a region of interest in the feature map, further analysed by a set of shared convolutional layers. Finally, the information extracted by the local branches and the main global branch are fused together for recognition. The experiments carried out on the challenging UBIRIS-v2 benchmark show that by integrating the proposed framework with various ResNet architectures, we consistently obtain an improvement in mAP of more than 4% over the “vanilla” architecture. In addition, extensive ablation studies were performed to better understand the behavior of the network and how the spatial transformation and the local branches influence the overall performance of the model. The proposed method can be easily adapted to other computer vision problems, which is also regarded as one of its strengths.

## 1. Introduction

Biometric identifiers refer to unique physical (iris, face, fingerprints, etc.), behavioural (gait, typing patterns), or physiological (EEG) traits that can be used to identify and describe individuals. The COVID-19 pandemic has caused a major decline in the performance of existing face identification systems [1], as a result of a large data drift. This is because many people wear protective masks that conceal most of the face area, leaving only the periocular region and the forehead visible.

Although there is no standardised definition from organisations such as NIST or ISO/IEC, the term periocular refers to the area around the eyes (i.e., the eyebrows, eyelashes, eye-folds, skin texture, tear ducts, etc.). Figure 1 shows an overview of a periocular re-identification system, in which the area surrounding the eye is used as a cue to determine the correct match between the query and the gallery individuals.

Ref [2], the periocular cues are categorised into (1) level one features, which are more dominant in nature and deal with geometry or shape (eyelids, eye corners, eyebrows), and (2) level two features, which are more related to colour and texture (skin appearance, skin pores, wrinkles, colour). Studies [3] have shown that the periocular region contains relevant cues for person identification both in visible (VIS) and near-infrared images (NIR), and that, in general, level two features are more efficient in VIS images, while level one features are useful in NIR images. Moreover, Ref. [4] showed that humans and computers rely on the same periocular cues for person identification, in both NIR and VIS scenarios.

In recent years, periocular recognition has become a prominent research area in biometric systems, because it has proved to be a valuable biometric approach and it offers several advantages. First of all, it can be captured with the same imaging devices used for facial or iris identification [3], it is non-intrusive and can be performed without the need for physical contact or cooperation from the subjects. In addition, periocular recognition can be performed in different spectra (both in visible-light and infrared spectrum), making it a versatile method of biometric identification.

All the cues present in the periocular area are prone to occlusions or other factors that influence their applicability:The iris and the sclera are sensitive to corneal reflexions (the Purkinje images);Accessories—such as eyeglasses or bangs—can occlude the eyebrows;Head/eye movements can lead to capturing blurry data;Makeup can influence the overall appearance of the eye.

With these challenges in mind, the key idea of this paper is to design a convolutional neural network that can adapt to these challenging environmental changes and automatically focus on the most relevant cues visible in the input image. To this end, we propose a multi-branch architecture that can focus on several bio-metrical traits at different granularities, by adding several local branches to focus on different regions of the image. Each local branch learns a geometrical transformation matrix (allowing for scaling and translation), which is then used to sample a region of interest (ROI) from an intermediate feature map. These ROIs are further analysed by a set of shared local layers, and then merged (via feature map summation) with the global branch. The workflow of the proposed solution is illustrated in Figure 2. By adding additional local branches to the model, we effectively exploit different bio-metrical traits and boost their performance in challenging environmental scenarios. Although the model design involves setting several hyperparameters, the features and the spatial transformation are automatically learned, and the proposed model can be trained in an end-to-end manner. Extensive ablation studies were performed to determine the impact of each hyperparameter.

The main contributions of this article are threefold: (1) we employ a spatial transformation module, which learns, in a semi-supervised manner, to identify the most prominent regions of the periocular area; (2) the periocular area is analysed both holistically, but also at the local level within the selected regions; and (3) the global and local information are fused to solve the recognition problem, and the loss function is applied to both the global and local branches to ensure that relevant features are extracted. Last but not least, the proposed method is generic, and can be easily adapted to other computer vision tasks.

The remainder of this manuscript is organized as follows: Section 2 reviews the existing approaches for periocular recognition, while the methodology used for conducting this study and the proposed method are introduced in Section 3. Section 4 presents the experimental results and a comparison with state-of-the-art works. The conclusions of the paper and several future research directions are summarized in Section 6.

## 2. Literature Survey

The pioneering work of Ref. [5] analysed the feasibility of periocular biometrics and proposed a system that exploits a set of global (LBP—Local Binary Patterns, HoGl—Histogram of Oriented Gradients) and local (SIFTl—Scale-Invariant Feature Transform) descriptors, fused together at the score level based on a weighted sum with min-max normalisation. Early works focused on designing effective feature descriptors to capture textual information around the eyes, either at a global level—(variants of) LBP [6,7], Gabor filters [8]—or at a local level—SIFT [9], SURF (Speeded Up Robust Features) [10], or SAFE (Symmetry Assessment by Feature Expansion) [11].

Global-based methods operate holistically on the entire periocular region and extract a feature vector based on texture or shape information. Several methods rely on LBP descriptors, which determine binary patterns by comparing each pixel with its neighbours. The global descriptor is then obtained by concatenating histograms of binary patterns computed across image cells. Ref. [7] investigated several feature extraction methods and determined that LBP substantially improved the performance of both verification and identification methods. In addition, they proposed the Local Walsh-Transform Binary Pattern feature representation, an effective variant of LBP. Other works use Gabor filters—with different orientations and frequencies—to analyse the texture in the periocular region.

A matching algorithm based on Gabor filters and a feature encoding scheme that relies on three operators to extract robust features in different spectral bands, was proposed in [8]. Two operators—Weber Local Descriptor (WLD) and uniform LBP—work on the magnitude of the filtered images, while another one—uniform generalised LBP operator (GLBP)—operates on the phase.

PPDM (Periocular Probabilistic Deformation Model) [12] applied a probabilistic inference model to compute 1:1 matching scores (between query and gallery images) based on correlation filters extracted from periocular image patches. Subsequently, the match performance was improved [13] with an unsupervised method used to select discriminate regions from the periocular area.

Local-based methods employ a multi-stage process: first, prominent keypoints are located within the periocular area, and then features are extracted from their vicinity. In Ref. [11], the authors adapted the Symmetry Assessment by Finite Expansion (SAFE) algorithm, previously used in fingerprint analysis, to the problem of periocular recognition. The key idea is to sample several keypoints based on a rectangular grid positioned in the eye centre, and then project ring-shaped areas of different sizes onto a space of harmonic functions used to determine symmetric curve families. A multi-modal authentication system that analyses and fuses features from the face, periocular and, if visible, iris area, is introduced in Ref. [10]. The system extracts three feature descriptors—SIFT, SURF, and Binarised Statistical Image Features (BSIF)—and explores various fusion strategies to effectively combine information from all three modalities. Ref. [14] identifies four prominent regions in the periocular area (eyebrows, upper eye fold, lower eye fold, and eye corners) and then computes a feature representation vector based on HOG, KAZE, and SING descriptors, as well as shape information. Finally, a Naïve Bayes classifier is used to perform the periocular recognition based on the extracted feature vector. The main drawback of this method is that it also requires the accurate segmentation of the features in the periocular area.

With the impressive advances of deep learning in the fields of computer vision and image recognition, recent developments in periocular biometrics focus exclusively on deep convolutional neural networks. Before CNNs, traditional pattern recognition methods used handcrafted features, such as LBP, HoG, Zernike moments [15], fast block processing feature extraction [16] etc., which were manually designed, taking into consideration the specific task and problem. On the other hand, in CNNs the features are automatically learned from the input data by convolutional layers. In general, deep learning methods tend to outperform traditional hand-crafted feature extraction techniques in computer vision tasks due to their ability to automatically learn relevant features from the data. The interested reader can refer to Refs. [3,17,18] for an in-depth presentation on early periocular research.

In Ref. [19], using transfer learning, seven CNN architectures were trained and compared in the context of periocular recognition. Ref. [20] proposed an original augmentation strategy based on multi-class region swapping, such that the network learns to consider the iris and the sclera regions as not reliable for biometric recognition, and only focus on the information surrounding the eyes. Although the method does not involve additional parameters or an increase in inference time, it completely disregards some regions in the periocular area that contain powerful biometric traits.

Other works [21,22] employed multi-task models to boost the performance of periocular recognition systems. Ref. [22] proposed semantics-assisted convolutional neural networks (SCNN) to incorporate explicit semantic information (gender and eye side): two separate CNNs are trained on these two tasks (identification and semantic task), and in the end are joined to obtain more powerful feature representations or to perform score fusion. Similarly, Ref. [21] introduced an end-to-end biometric system, based on a multi-task architecture. The framework features a shared convolutional backbone and two separate, dedicated branches, one for biometric identification and one for soft biometric recognition. These branches are fused together for the final periocular recognition, while also predicting soft biometrics. However, the main disadvantage of these approaches is the need for annotated datasets with identity and soft biometric attributes, which are not always available. In addition, they tend to involve more parameters for the supporting tasks.

In Ref. [23], the authors proposed a two-branch deep learning model to analyse iris and periocular cues, and then fuse the corresponding predictions through a multilayer perceptron (MLP). The training procedure is rather complicated, as it requires several stages. In addition, the inputs to each branch require different pre-processing techniques and, as two CNNs are used, the system has more learnable parameters, which leads to longer inference times.

Ref. [24] designed a multimodal biometric system to exploit facial and periocular cues. The proposed model features a shared convolutional backbone, as well as two predictor branches to accommodate the two modalities. During training, additional loss functions are defined to decrease the distance between feature embeddings or periocular-face intra-subjects, while simultaneously maximising feature embeddings of the periocular-face inter-subjects. In Ref. [25] the authors proposed AttenMidNet, a lightweight CNN based on attention mechanisms. The building blocks of the architecture are the MCRS blocks that comprise a convolutional layer, a squeeze-and-excitation block [26], and a residual connection. Ref. [27] proposed a Siamese-like dual stream network, which analyses in parallel the left and right periocular regions of a subject, and then investigates the feature aggregation techniques of the two streams.

Table 1 provides a summary (features, highlight, brief methodology) of the related periocular recognition methods discussed in this section.

Despite their impressive performance, a major caveat of deep learning methods is their lack of explainability. As a result, some works [29,30] tackled the problem of visual explanations and interpretable artificial intelligence in the context of periocular recognition.

## 3. Materials and Methods

### 3.1. Problem Setting

We formulate the biometric recognition problem as a re-identification problem. Consider a training image-based set of *N* different identities, each containing $n_{id}$ samples. The purpose of a re-identification system is to learn a function that will find the best match between a query image and a gallery set of images. The query set contains images of the periocular area of a subject we want to identify in another image or set of images (the gallery set). The gallery set contains the potential matches (periocular images) for the target person in the query set.

### 3.2. Solution Outline

The periocular area comprises various anatomical cues suitable for recognizing individuals and numerous studies have analysed their significance for this process (refer to Ref. [3] for a detailed survey on this matter). However, the applicability of the periocular biometric traits is influenced by environmental factors, and depending on the overall appearance of the eye area and the image capture modality (VIS or NIR), one cue might be more relevant than the other. As an example, the skin texture and the overall eye shape are suitable cues for VIS images [4], but make-up can influence their appearance and therefore degrade the performance of a machine learning model that has been trained to focus on such features. With this in mind, we devised a deep learning model which uses several local branches trained to locate (in a semi-supervised manner) and analyse several discriminative regions in the periocular area (Figure 2).

The proposed method can be easily integrated into any network architecture, and it can effectively boost the performance of (re-)identification systems with a small increase in inference time. Additionally, this strategy can be easily adapted to other image recognition tasks.

### 3.3. Model Architecture

The key idea of our method is to employ *L* branches, branched from the *i*th level of a neural network architecture, which will learn to extract prominent ROIs within the periocular area and analyse them for biometric identification. Each local branch starts with a *Spatial transformation*module, responsible for the selection of an ROI in the input feature map $f_{i}$. Then, a set of *B* shared convolutional layers (between all the *L* branches) process the selected areas, and finally, their corresponding feature maps are fused through element-wise summation. This summation result is also added to the global branch. To ensure that the local branches actually learn relevant information, we apply the loss function to both the global branch’s output and the summation of the local branches.

#### 3.3.1. The Local Branch

The local branches learn to spot the most relevant regions on the input feature map and attempt to solve the identification problem based solely on the features from these areas.

Inspired by Refs. [31,32], we employ a visual attention mechanism to locate the discriminative parts of a feature map $f_{i}\in R^{h\times w\times x}$. To achieve this, each local branch starts with a *Spatial transformation* module, which learns to select an ROI from the input feature map. Similar to [33], we generate a spatial map using grouping operations to compute two bi-dimensional maps: $f_{avg}\in R^{h\times w\times 1}$ and $f_{max}\in R^{h\times w\times 1}$ and aggregate them into a single map $s_{i}$ using element-wise summation. $s_{i}$ is then passed to a feed-forward multi-layer perceptron that regresses an affine spatial transformation matrix θ:(1)$$\theta =\left[\begin{array}{ccc} s_{x} & 0 & t_{x}\\ 0 & s_{y} & t_{y} \end{array}\right].$$

This matrix allows for cropping and translation (2D spatial parameters $t_{x}$ and $t_{y}$) and image scaling (scaling parameters $s_{x}$ and $s_{y}$). The transformation is not learned explicitly from the dataset labels; instead, the model automatically optimises these parameters such that it boosts the recognition accuracy. At the beginning of the training process, the weight and biases of the linear layer are initialised with the identity transformation (i.e., all weights initialised to 0, biases for $s_{x}$ and $s_{y}$ initialised to 1, biases for $t_{x}$ and $t_{y}$ set to 0). After the transformation matrix θ (Equation (Equation 1)) is determined, the *grid generator* module computes a 2D flow-field grid based on θ to generate the coordinates from the input image corresponding to each position in the output. The *grid sampling* module applies the transformation parameters to the input and returns an ROI $r_{i}$ from the input feature map. The structure of this *Spatial transformation* module is depicted in Figure 3, and, for clarity, is also detailed in Algorithm 1. As illustrated in Figure 2, the selected ROI $r_{i}$ is further passed through a set of shared convolutional layers. In the end, all the outputs of the local branches are brought to agree with the shapes via an extrapolation layer, and they are added to the output of the global branch.

To guarantee that the features learned by the local branches are in fact useful in the periocular identification task, the loss function is also applied to their summation. More precisely, during training, the model has multiple outputs, one corresponding to the global branch and one corresponding to the summation of the local branches’ output. However, at test time, only the global branch output is used and evaluated. This is inspired by Ref. [34], where the authors used several small classifiers (discarded at test time) on top of come convolutional blocks to ensure that the layers in the middle of the network are also very discriminative.


**Algorithm 1:** Spatial transformation module. **Input**: $f_{i}$ - feature map. **Output**: ${f\prime }_{i}$ - ROI in $f_{i}$ /* compute μ channel-wise mean                                   */ $\mu \leftarrow mean_{ch}\left(f_{i}\right)$; /* compute *m* channel-wise maximum                                */ $m\leftarrow max_{ch}\left(f_{i}\right)$; /* fuse(addition) μ and *m* and flatten result                     */ p← flatten(μ+m); /* apply multi-layer perceptron to compute transformation matrix θ( */ θ←MLP(p) /* generate affine grid based on θ and sample $f\prime_{i}$                */ $f\prime_{i}\leftarrow$ sample($f_{i},\theta$); **return**
$f\prime_{i}$


#### 3.3.2. Closed vs. Open-World Operating Modes

In the context of periocular identification, AI models can operate in closed or open-world modes, depending on whether the identities of the subjects to be recognised are known or not.

In the closed-world setting (i.e., when all the subjects are known in advance), the identification problem can be formulated as a classification task, and a softmax layer can be used to predict the identities. In this case, the final layer of CNN is a dense layer with softmax activation, and its number of neurons is equal to the number of identities in the dataset. This problem can be seen as a watch-list identification problem [20], in which the model aims to spot some subjects from a predefined list.

On the other hand, in the open-world setting, when the set of identities is unknown, the model needs to be trained to distinguish between unseen subjects. In this case, the identification problems are formulated as a distance learning task or a retrieval ranking problem. Therefore, the learning process aims to encode an input image into an embedding space, such that the distance between the images of the same identity is small, while the distance between images from different identities is large. Consequently, the final layer of the model is used as a feature descriptor (and not as a classification layer as in the closed-world setting). For this setup we employed the triplet loss (Equation 3) to optimize the model. Once the model is trained, the recognition and verification task becomes straightforward in the embedding space, as it simply involves the computation between the computed embeddings. The forward pass of the proposed network architecture is illustrated in Algorithm 2.

We evaluated the proposed method for both closed-world and open-world settings.

### 3.4. Training Process

The proposed method is trained in an end-to-end manner. As mentioned above, biometric identification systems can operate in either *closed-world* or *open-world* settings.

In closed-world settings, test subjects are known at train time, and the identification task becomes essentially a classification problem. For this test setup, the identification loss is the standard categorical cross entropy loss: $L_{ID}\left(\hat{x},y\right)=-\sum y\cdot log\left(\zeta \left(\hat{x}\right)\right)$, where *y* is the ground truth identity for the *x* sample, $\hat{x}$ is the model prediction (logits), and ζ is the softmax function: $\zeta \left(z_{i}\right)=\frac{e^{z_{i}}}{\sum_{j}e^{z_{j}}}$.

During training for closed-world scenarios, the model has two outputs: $\hat{x_{g}}$, the response of the classification layer in the global branch, and $\hat{x_{l}}$, the response of the classification layer applied on the summation of the feature maps from the local branches. The final loss function $L_{CW}$ for the closed-world setting is given in Equation (Equation 2):(2)$$L_{CW}=L_{ID}\left(\hat{x_{g}},y\right)+\sum_{l=1}^{L}L_{ID}\left(\hat{x_{l}},y\right).$$

On the other hand, in open-world scenarios, the output of the network does not consist of class probabilities, but in a feature vector in a lower dimensional embedding space in which the L2 distances correspond to subject similarities. In this case, a variant [35] of the triplet loss function [36] is used:(3)$$L_{T}=\sum_{\begin{array}{c} a,p,n\\ y_{a}=y_{p}!=y_{n} \end{array}}\max(||f\left(x_{a}\right)-f\left(x_{p}\right){\left|\right|}^{2}-||f\left(x_{a}\right)f\left(x_{n}\right){\left|\right|}^{2}+\lambda ,0)).$$

The triplet loss ensures that given an anchor point $x_{a}$, the projection of a positive sample $x_{p}$ (belonging to the identity $y_{a}$) is closer to the anchor’s projection than that of a negative sample of a different identity $y_{n}$, by at least a margin of λ. In addition, during training, we also apply a final classification layer with the number of neurons equal to the number of identities in the training set, on top of the global branch $cl_{g}$, and on top of each local branch $cl_{l},l\in \left\{1,...,L\right\}$.

To sum up, the loss function for the open-world setting is specified in Equation (Equation 4):(4)$$L_{OW}=\omega_{1}(L_{T}\left(x_{g}\right)+\sum_{l=1}^{L}\left(L_{T}\left(x_{l}\right)\right)+\omega_{2}(L_{ID}\left(cl_{g}\right)+\sum_{l=1}^{L}\left(L_{ID}\left(cl_{l}\right)\right),$$
where $\omega_{1}$ and $\omega_{2}$ are the weights for the triplet and identification losses. In our experiments we set $\omega_{1}=1$ and $\omega_{2}=1$.

All models were trained using transfer learning (using pre-trained weights from ImageNet—from python torch library version 1.13.0 [37]) for 70 epochs, using the Adam optimiser, with an initial learning rate of 0.015, updated with a step decay scheduler at epochs 25 and 50.


**Algorithm 2:** Forward function of the proposed network architecture.

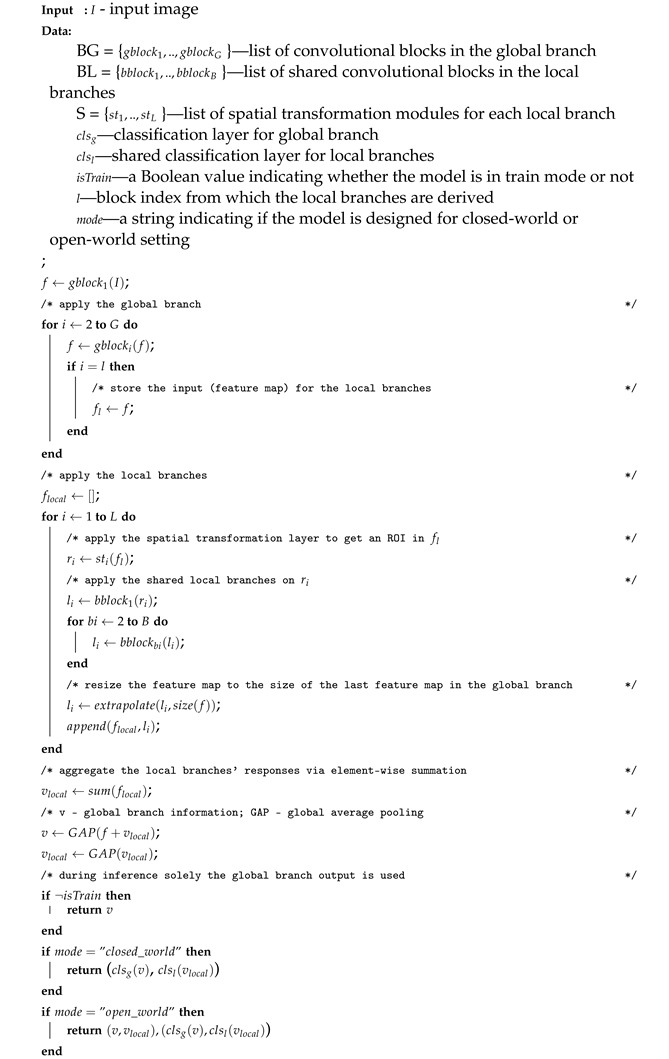




## 4. Results

This section presents the experiments performed for evaluating the performance of the proposed method for periocular biometrics, following the methodology introduced in Section 3.

### 4.1. Experimental Setup

The proposed method can be incorporated into any network architecture, but we chose the ResNet architecture [38] due to its widespread use in computer vision tasks. The key feature of this architecture is the use of residual connections, which allow the layers of the model to learn a residual mapping H(x)=x+F(x), instead of directly fitting a function F(x). This is achieved by defining convolutional blocks (Figure 4) with shortcut connections (skipping one or more layers), which simply perform an identity mapping by adding the input of one convolutional block to its output. With this strategy, it is ensured that the network can learn an identity function when the residual connections are not required, and therefore much deeper architectures can be created.

The ResNet architecture comprises a chain of such blocks with residual connections and its generation follows some simple rules: the convolutional layers in the ResNet blocks have a size of 3×3; the depth of a layer (number of convolutional filters) is the same if the feature maps have the same spatial size; and when the spatial size of the feature maps is halved, the number of filters of the layer is doubled. Several architectures [38] with different depths have been defined, depending on the number of blocks chained together. In this paper, we experimented with ResNet-18 and ResNet-34 architectures (Table 2).

The number of local branches used for these experiments is B=3, and they are added from the 13th layer of the network.

The proposed method can be easily adapted for other network architecture, as it involves only adding the local branches on top of a feature map extracted from some layer in the network.

### 4.2. UBIRIS-v2 Dataset

UBIRIS-v2 [39] is a periocular image dataset designed to evaluate iris recognition models in challenging environments. The dataset contains 11,102 periocular images corresponding to the regions of the left and right eye of 261 subjects. In addition, each image in the dataset is also annotated with the distance to the camera, and the subject’s gaze. We have used this dataset to evaluate the performance of our model in open-world and closed-world settings.

### 4.3. Results

#### 4.3.1. Closed-World Setting

For the closed-world setting—i.e., when the test identities are known in advance—we employed the following data split protocol: the dataset is divided into two disjoint subsets, such that 80% of the samples are used in the training process, and the remaining 20% are used for performance evaluation. The splits were then automatically validated to check that they contain images belonging to all of the subjects in the dataset, and thus the closed-world assumption is satisfied.

Figure 5 illustrates the evolution of the metrics in the training process for the closed-world setting.

In Table 3, ResNet18 (baseline) refers to a ResNet18 architecture trained based on the schedule described in Section 3.4 and ResNet18 + local is a ResNet-18 architecture with 3 local branches added after the 13th convolutional layer. In terms of Rank-1, the proposed method is almost identical to the ResNet-18 baseline, but since the Rank-1 of the baseline is close to 100%, it is difficult to achieve notable improvements. On the other hand, the mAP of the proposed method surpasses the baseline by 10.15%.

The proposed method surpasses other periocular recognition methods by a considerable amount. This is due to the fact that by also training the local branches in the network to solve the periocular recognition problem based on the selected periocular regions, the discriminative power of the network is increased and therefore it improves the overall periocular recognition performance.

However, a direct numerical comparison is not completely fair, as other methods use different network architectures and training setups. Both SCNN [22] and ADPR [21] rely on multi-task learning models and these methods require labelled data not only for identification but also for auxiliary tasks. The additional classification streams added in these networks seem to have a high impact on the classification performance; in Ref. [21] if solely the periocular recognition stream is branch, without the attribute classification branch, the Rank-1 drops from 92.68% to 83.95% (with 8.7%). In addition, the training procedure of Ref. [21] is not performed in an end-to-end manner, and it involves several training steps. This is a cumbersome process, as at each step the appropriate optimisation hyperparameters need to be found, and the overall training takes longer. DEEPPRWIS [20] relies on an original augmentation strategy used solely during the training process, in which the model is automatically trained to focus on the area outside the eye (i.e., ignoring the iris and the sclera). One reason for the lower performance of DEEPPRWIS is that in many images in the UBIRISv2 dataset the area outside the eye is barely visible and the images are misaligned. As the key idea of DEEPPRWIS is to rely on the outside eye area, it is expected to have a lower performance in such cases.

#### 4.3.2. Open-World Setting

For the open-world setting, we selected the first 19 subjects as test samples, and the rest of the images (belonging to 142 subjects) were used to train the model. To avoid a friendly split of the dataset, and inspired by the problem of person re-identification [40], we considered the gaze of each subject when splitting the images into sets and gallery sets.

Figure 6 illustrates the evolution of the metrics over the training process for the open world setting.

Table 4 reports the results in the open-world setting. ResNet18 (baseline) and Resnet34 (baseline) refer to the "vanilla" Resnet architectures trained for the problem of iris re-identification.

The proposed method exceeds the baselines by a large amount in terms of both mAP and Rank-1. For the ResNet18 architecture, the mAP improved from 66.08% to 72.65% (with 6.57%) and the gain in Rank-1 is 2.57%. For the ResNet-34 architecture, the improvements are slightly lower: 3.22% in mAP and 1.29 in Rank-1.

The performance boost is more visible in the open-world scenario. In this case, the baseline ResNet model has lower accuracy allowing for greater improvement, both in terms of mAP and Rank-1, as opposed to the closed-world setting where the Rank-1 is saturated at around 99%. In the open-world scenario, the model learns to project the images into a lower-dimensional embedding space in which samples belonging to the same identity are close together, while the ones belonging to different identities have larger distances. Another possible explanation resides in the fact that in such scenarios, the fine-grained details extracted by the local branches are more useful in the recognition process.

## 5. Ablation Studies

The idea of this paper is to add additional local branches to a neural network architecture, which will learn to identify and analyse the most relevant regions in the input. There are several hyper-parameters that can influence the effectiveness of the proposed method, and, in this section, we will analyse their effect on the model’s performance. All experiments for this ablation study were performed using the ResNet-18 architecture.

The first hyper-parameter is the depth (layer number) at which the local branches are added to the model. In Table 5, the column *depth* indicates the value of this parameter. As discussed in Section 4.1, ResNet-18 architecture comprises a convolutional layer, followed by four types of ResNet block, each repeated twice (Table 2). In other words, the local branches were added before the second, third, and fourth ResNet block pairs of the ResNet-18 architecture. The results indicate that better results are obtained when the local branches are added deeper into the network architecture. This is somewhat expected, as deeper layers operate with more semantically meaningful features. In addition, this setup allows the model to use the shallower layers as a shared local feature extractor, and therefore to reduce the inference time and learnable parameters.

Another experiment we performed is related to the way the feature map from the global branch is being preprocessed by the *Spatial transformation* module (Figure 3). To this end, we experimented with three strategies: *Bottleneck*, *Channel*, and *Spatial*.

For the *Bottleneck* setup (in Table 5, *Bottleneck* in the column *Preprocessing*), the feature map is first passed through a 1×1 convolutional layer to reduce its depth to ch=32, thus making the computations more feasible (Equation (5)). Next, the processed feature map ${f\prime }_{i}$ is flattened and then processed by the MLP to compute the transformation matrix θ.
(5)$${f\prime }_{i}=conv_{1\times 1}^{32}f_{i},$$
where $conv_{1\times 1}^{32}$ denotes a 1×1 convolutional layer with 32 filters.

The results show that this strategy achieves the lowest performance—even lower than the baseline ResNet18 architecture. The MLP layer needs to process a high dimensional ($h_{f_{i}}\times w_{f_{i}}\times 32$, where $h_{f_{i}}\times w_{f_{i}}$ represents the spatial size of the feature map) flattened feature vector. This can lead to optimisation issues (overfitting) as the input layer in the MLP has a large number of neurons. Moreover, local branches might fail to extract semantically meaningful information from the flattened input feature.

Another strategy is to use global average pooling operations as a preprocessing step, as given in Equation (5): (6)$${f\prime }_{i}=gap\left(f_{i}\right)+gmp\left(f_{i}\right),$$
where gap and gmp represent the global average pooling and global max pooling operators, respectively.

This is similar to the Channel Attention from Ref. [33]. In this case, the input size of the MLP layer is $1\times 1\times c_{f_{i}}$, where $c_{f_{i}}$ is the number of channels in the input feature map and this vector models the information about the channels with the most prominent features responses. With this setup, when local branches are added to deeper layers of the network, the baseline ResNet-18 architecture is surpassed in both mAP and Rank-1.

However, the best result is obtained with the *Spatial* preprocessing (Equation (Equation 7)). For this setup, the MLP takes as input the sum between the channel-wise average and maximum pooling operations, similar to the spatial attention component from Ref. [33]:(7)$${f\prime }_{i}=cap\left(f_{i}\right)+cmp\left(f_{i}\right),$$
where cap and cmp represent the channel-wise average pooling and max pooling, respectively. With this configuration, the proposed method surpasses the baseline ResNet-18 architecture with more than 6% in mAP and 2.56% in Rank-1. Intuitively, this seems to be the most promising choice, since the goal of the *Spatial transformation* module is to select the most discriminative periocular areas. Equation (Equation 7) returns a map of dimensions $h_{f_{i}}\times w_{f_{i}}$, in which higher values indicate areas with prominent features.

## 6. Conclusions

This paper presented a periocular recognition system suitable for VIS images based on convolutional neural networks. The main contribution is to equip (a generic) model architecture with additional local branches that will learn, in a semi-supervised manner, to focus on the most discriminative features around the eyes. Unlike existing models, the proposed model is able to extract and exploit multiple regions in the periocular area in a lightweight end-to-end network architecture. Extensive experiments and ablation studies, both for closed-world and open-world setups, prove the effectiveness of the proposed solution. The limitations of the current work are related to the additional hyperparameters that need to be fine-tuned, and also to an increase in the number of learnable parameters.

In future work, we plan to experiment with other strategies for combining the response of the global and the local branches, such as more complex ensembling methods or attention mechanisms.

## Figures and Tables

**Figure 1 sensors-23-02456-f001:**
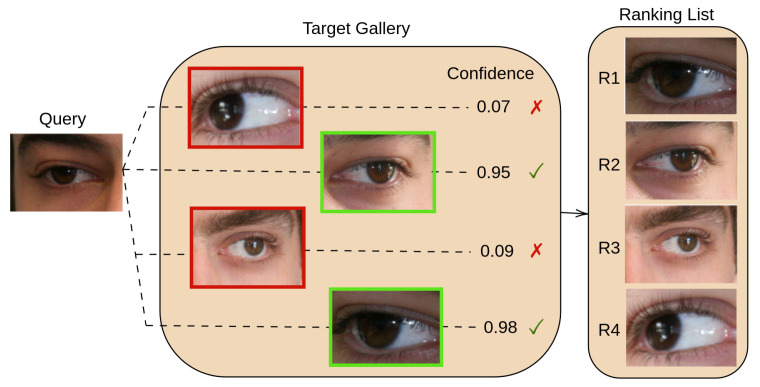
Periocular re-identification system. The goal is to find the most similar correct matches of a query person in the target gallery. R stands for rank, and confidence is the prediction probability of the biometric system.

**Figure 2 sensors-23-02456-f002:**
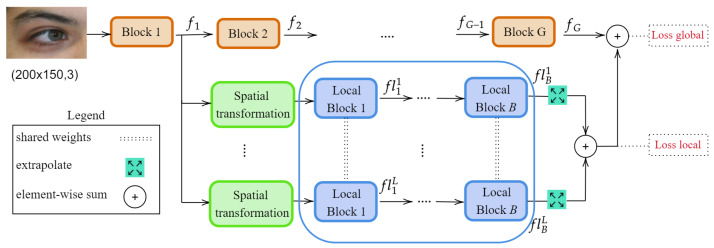
The overview of the proposed deep learning model based on convolutional neural networks. The input image is processed using several global and local convolutional blocks. *Block* and *Local Block* refer to the convolutional blocks/layers in the global branch (depicted in orange) and local branch (depicted in blue), respectively. The feature map $f_{i}$ from some level *i* in the network architecture is fed to L≥1 local branches. Each branch applies a *Spatial transformation* module to select an ROI within the input feature map, and then further analyses this region by applying several *shared* convolutional blocks. In the end, the feature maps of the local branches are extrapolated to the same size and fused with the global branch via element-wise summation. To ensure that the local branches extract useful features, the loss function is applied to both the global branch and the summation of the local branches.

**Figure 3 sensors-23-02456-f003:**
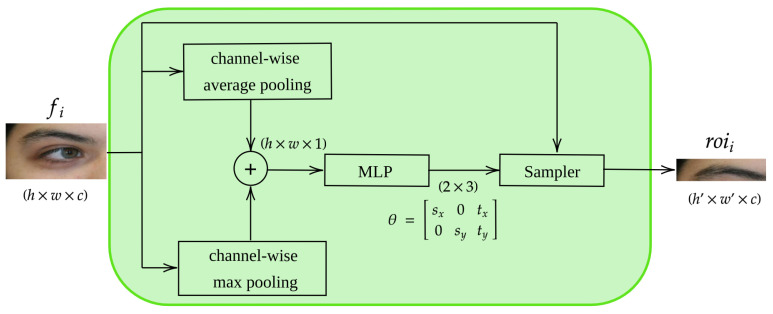
Detailed architecture of the *Spatial transformation* module. The input convolutional feature map $f_{i}$ first undergoes two channel-wise pooling operations (maximum and average) to highlight the most discriminative spatial areas. Their outputs are then added together and passed to a multi-layer perceptron to learn θ - the transformation matrix. θ is used to sample a region of interest from $f_{i}$, which will be processed by the local branches. Although the *Spatial transformation* module operates on intermediate feature maps, for illustration purposes we exemplify its function with an image (corresponding to the input feature map).

**Figure 4 sensors-23-02456-f004:**
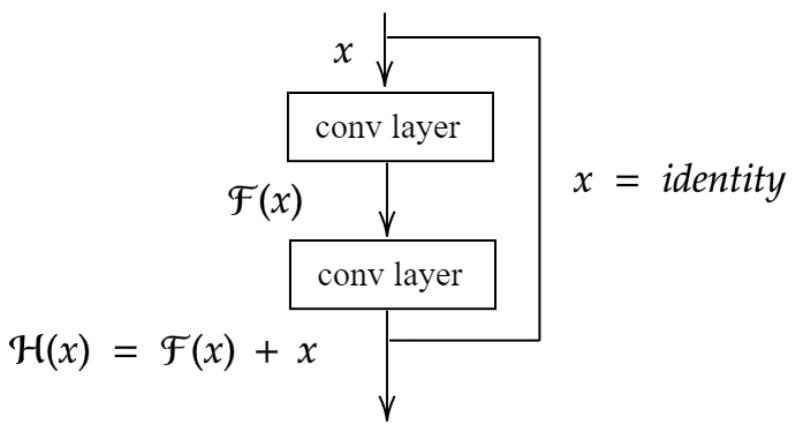
The structure of a ResNet block, adapted from Ref. [38].

**Figure 5 sensors-23-02456-f005:**
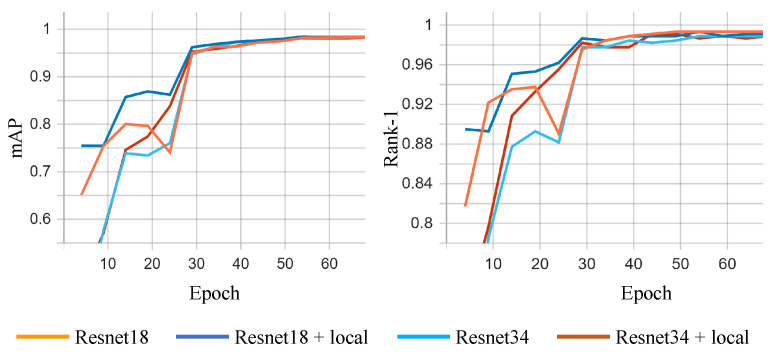
Metrics mAP (mean Average Precision) and rank-1 evolution during the training process for the closed-world setting.

**Figure 6 sensors-23-02456-f006:**
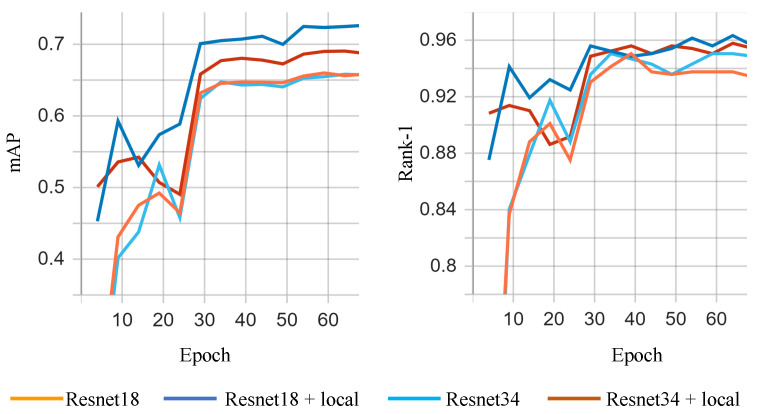
Metrics (mAP and rank-1) evolution during the training process for the open world setting.

**Table 1 sensors-23-02456-t001:** Representative research papers on periocular recognition. As in the survey papers [3,18], the methods are grouped based on the feature extraction strategy.

Strategy	Reference	Highlight
Global features	[5]	Global features: LBP + HoG, fused together at score level
	[12]	probabilistic inference model based on correlation filters
	[8]	Gabore filters + feature encoding scheme to extract features in different spectral bands
Local features	ISURE [9]	Multinomial Naïve Bayes learning + Dense SIFT for nearest neighbour matching
	[11]	Symmetry Assessment by Finite Expansion for periocular recognition
	[10]	multimodal system (face, periocular, iris) based on SIFT, SURF, and BSIF fusion
	[14]	HOG, KAZE, SING features from four regions in the periocular area + Naïve Bayes classifier
Deep learning	SCNN [22]	several CNNs to incorporate semantic information (gender and eye type) in the training process
	DEEPRWIS [20]	augmentation strategy to focus solely on the outside eye areas
	[19]	transfer learning on different CNN architectures
	ADPR [21]	multitask CNN for periocular recognition and soft attribute prediction
	AttenMidNet [25]	attention mechanism (squeeze-and-excitation blocks) and mid-level features
	Dual-Input CNN [28]	dual-stream Siamese-like CNN + fusion scheme

**Table 2 sensors-23-02456-t002:** Resnet architectures.

Network	Architecture				
ResNet-18	(7×7,64)/2	$2\times \left(\begin{array}{c} 3\times 3,64\\ 3\times 3,64 \end{array}\right)$	$2\times \left(\begin{array}{c} 3\times 3,128\\ 3\times 3,128 \end{array}\right)$	$2\times \left(\begin{array}{c} 3\times 3,256\\ 3\times 3,256 \end{array}\right)$	$2\times \left(\begin{array}{c} 3\times 3,512\\ 3\times 3,512 \end{array}\right)$
ResNet-34	(7×7,64)/2	$3\times \left(\begin{array}{c} 3\times 3,64\\ 3\times 3,64 \end{array}\right)$	$4\times \left(\begin{array}{c} 3\times 3,128\\ 3\times 3,128 \end{array}\right)$	$6\times \left(\begin{array}{c} 3\times 3,256\\ 3\times 3,256 \end{array}\right)$	$3\times \left(\begin{array}{c} 3\times 3,512\\ 3\times 3,512 \end{array}\right)$

**Table 3 sensors-23-02456-t003:** Recognition results on UBIRIS-v2 dataset for the closed-world setting. The best performing models are highlighted in **bold**.

Method	mAP	Rank-1
SCNN [22]	-	79.30%
DEEPPRWIS [20]	-	87.64%
ADPR (PR) [21]	-	83.95%
ADPR (JPR )[21]	-	92.68%
ResNet18 (baseline)	87.81%	99.33%
ResNet18 + local (**ours**)	**97.96**%	**99.33**%
ResNet34 (baseline)	95.81%	98.88%
ResNet34 + local (**ours**)	**96.47**%	**99.10**%

**Table 4 sensors-23-02456-t004:** Recognition results in the UBIRIS-v2 dataset for the open-world setting. The best performing models are highlighted in **bold**.

Method	mAP	Rank-1
ResNet18 (baseline)	66.08%	93.02%
ResNet18 + local (**ours**)	**72.65**%	**95.59**%
ResNet34 (baseline)	65.81%	95.04%
ResNet34 + local (**ours**)	**69.03**%	**96.33**%

**Table 5 sensors-23-02456-t005:** Ablation study of the depth at which local branches are added to the network and the strategy for preprocessing the global branch feature map.

Method	Preprocessing	Depth	mAP	Rank-1
ResNet18(baseline)	–	–	66.08%	93.03%
ResNet18 + local	Bottleneck	5	61.66%	93.76%
ResNet18 + local	Bottleneck	9	60.38%	93.76%
ResNet18 + local	Bottleneck	13	61.50%	93.76%
ResNet18 + local	Channel	5	62.64%	95.04%
ResNet18 + local	Channel	9	68.78%	96.14%
ResNet18 + local	Channel	13	71.19%	95.22%
ResNet18 + local	Spatial	5	61.26%	93.57%
ResNet18 + local	Spatial	9	66.09%	94.86%
ResNet18 + local	Spatial	13	72.64%	95.59%

## Data Availability

In this article, the publicly available UBIRIS.v2 dataset was used: http://iris.di.ubi.pt/ubiris2.html (accessed on 29 January 2023). The starting code for this work is from Ref. [41].

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
