# Peer review of "Adaptive Spatial Transformation Networks for Periocular Recognition"

_sensors, 2023, doi:10.3390/s23052456_

Round 1

Reviewer 1 Report

1) The literature survey should critically analyse earlier works than just summarizing the previous works.

2) In section 3.1.1, the authors claim that they analyse both closed world settings and open world settings. But they haven't say anything about open world settings. How do you implement it? If unseen classes, is it something like unsupervised approach? 

3) The abstract talks about some deep learning architectures. But , it seems the features are extracted by the user in the methodology part. It seems slightly confusing and contradictory since deep learning has automated feature extraction. Clarify it.

4) Where do you use deep learning/its characteristics in the proposed method?

5)  The results are given theorotically and only the final results are given. It should include more detailed results with tables. Confusion matrix can eb included. Training curves can be included.

6) what is the need for ablation studies in this work? How do you correlate it with the objectives given in the abstract?

Reviewer 2 Report

A periocular recognition framework based on deep learning architectures is proposed. The proposed approach is theoretically sound. The performance of the proposed approach is experimentally evaluated for open world and closed world setting and shows improvement over the existing approaches.
It is not clear why Resnet-34 is not considered for the closed world setting in Table 2. This can be added to the experimental results. Overall, the manuscript is well presented and makes a good contribution.

Reviewer 3 Report

Summary:

In this work, a periocular recognition framework based on deep learning architectures is presented. In this work, the most important areas in the periocular region are localized and analyzed. Several local branches to focus on different regions of the image. Each local branch learns a geometrical transformation matrix. Then, it is used to sample a region of interest from an intermediate feature map. The ROIs are further analyzed by a set of shared local layers and merged with the global branch.

The manuscript is interested; however, the following comments need to be addressed :

Comments :

Abstract:

1 – The authors should use either American or Britain English. For example, “localized” and “analysed”.

2 – The abstract will be self-contained if the authors included the results in terms of improvement ratio between the presented work and existing works in the abstract.

Introduction Section:

3 – In line 59, referencing to Figure is incorrect. Please check and correct.

4 – The Introduction section is very concise. The author may include some feature types. For example the difference between hand crafted features (a- doi: 10.1016/j.patcog.2016.02.014, and b- doi: 10.3390/sym14040715 ) and convolutional neural network.

5 – The contributions need to be included as a list at the end of this section (before the manuscript organization paragraph).

Literature survey Section:

6 – A summary table of the literature survey would be nice if included. The summary table includes, for each related work: a) algorithm name and reference, b) brief methodology, c) highlights, and d) limitations.

Materials and Methods Section:

7 – References for Equations should be included.

8 – It would be more readable to the readers when a pseudo code is included. The pseudo code can be divided into multiple parts for more interpretation, i.e., pseudo code for each part can be included.

Results Section:

9 – This section requires more discussion and analysis. Also, more recent works should be included in the comparison.

Conclusions Section:

10 – Include limitations for the presented work.

References:

11 – More recent works from 2021 and 2022 need to be included.

Round 2

Reviewer 1 Report

It can be accepted now 

Reviewer 3 Report

Summary:

In this work, a periocular recognition framework based on deep learning architectures is presented. In this work, the most important areas in the periocular region are localized and analyzed. Several local branches to focus on different regions of the image. Each local branch learns a geometrical transformation matrix. Then, it is used to sample a region of interest from an intermediate feature map. The ROIs are further analyzed by a set of shared local layers and merged with the global branch.

The authors have addressed the raised comments.